# Full-color persistent room temperature phosphorescent elastomers with robust optical properties

Juan Wei[1], Mingye Zhu[1], Tingchen Du[1], Jangang Li[1], Peiling Dai[1], Chenyuan Liu[1], Jiayu Duan[1], Shujuan Liu[1], Xingcheng Zhou[1], Sudi Zhang[1], Luo Guo[1], Hao Wang[1], Yun Ma ®[1] ✉, Wei Huang ®[1,2] ✉ & Qiang Zhao ®[1,3] ✉

Persistent room temperature phosphorescent materials with unique mechanical properties and robust optical properties have great potential in flexible electronics and photonics. However, developing such materials remains a formidable challenge. Here, we present highly stretchable, light-weight, and multicolored persistent luminescence elastomers, produced by incorporating ionic room temperature phosphorescent polymers and poly-vinyl alcohol into a polydimethylsiloxane matrix. These prepared elastomers exhibit high optical transparency in daylight and emit bright persistent luminescence after the removal of 365 nm excitation. The homogeneous distribution of polymers within the matrix has been confirmed by confocal fluorescence microscopy, scanning electron microscopy, and atomic force microscopy. Mechanical property investigations revealed that the prepared persistent luminescence elastomers possess satisfactory stretchability. Impressively, these elastomers maintain robust optical properties even under extensive and repeated mechanical deformations, a characteristic previously unprecedented. These fantastic features make these persistent luminescence elastomers ideal candidates for potential applications in wearable devices, flexible displays, and anti-counterfeiting.

Long-afterglow luminescence, an appealing optical phenomenon that lasts for at least several seconds after the removal of the excitation source[1,2], has garnered considerable attention due to its vast potential in applications such as bioimaging[3,4], display[5,6], sensor[7,8], and anti-counterfeiting[9–12]. In addition to conventional inorganic materials, organic ultralong room temperature phosphorescent (RTP) materials have also been extensively investigated in recent years, due to their high biocompatibility, low cost, and easy molecular structure

modification[13–15]. To date, numerous organic persistent RTP materials with high quantum yields and ultralong long emission lifetimes have been reported[16–20]. However, most of these are limited to molecular crystals[21–23], which severely constrain their practical applications because of the stringent conditions required for cultivating the crystalline state. In this context, the achievement of organic persistent RTP polymers with good flexibility, large-area processability, and producibility were developed by incorporating small-molecule phosphors

[1]State Key Laboratory of Organic Electronics and Information Displays & Jiangsu Key Laboratory for Biosensors, Institute of Advanced Materials (IAM) & Institute of Flexible Electronics (Future Technology), Nanjing University of Posts and Telecommunications (NUPT), Nanjing 210023, P. R. China. [2]Frontiers Science Center for Flexible Electronics (FSCFE), MIIT Key Laboratory of Flexible Electronics (KLFE), Northwestern Polytechnical University, Xi'an 710072, P.R. China. [3]College of Electronic and Optical Engineering & College of Flexible Electronics (Future Technology), Jiangsu Province Engineering Research Center for Fabrication and Application of Special Optical Fiber Materials and Devices, Nanjing University of Posts and Telecommunications (NUPT), 9 Wenyuan Road, Nanjing 210023, P.R. China. ✉e-mail: iamyma@njupt.edu.cn; vc@nwpu.edu.cn; iamqzhao@njupt.edu.cn

into a commercially available polymer matrix. This approach is one of the most popular methods for improving the processability of persistent RTP materials[20–23]. Poly(methylmethacrylate) (PMMA) and polyvinyl alcohol (PVA) are the most commonly used polymer matrices[24–27], but their stretchability is poor.

Polymer composites with a combination of high-efficiency luminescence and good elastomeric properties are promising in wearable optoelectronic devices, flexible displays, and anti-counterfeiting[28–31]. Recently, persistent RTP elastomers have been developed by doping small-molecule phosphors into thermoplastic polymer matrices[32,33]. Nevertheless, upon application of drawing force, their luminescence intensity and emission lifetime in the deformation region were remarkably decreased or even quenched. The constant mechanical deformations in practical applications make those materials dissatisfy the requirements for real-world usage. This is because the compact interactions between chromophores and polymer matrix, as well as interactions among chromophores themselves, would be broken under the large-area deformation. Therefore, it is highly desirable to develop persistent RTP materials that can maintain their optical properties in the deformation region and after multiple-time bending and stretching. Alternatively, blending amorphous RTP copolymers into elastomer matrices might be an effective way to obtain stretchable films with stable persistent luminescence properties. This is because the long polymer chains of RTP copolymers can resist external stretch to some extent (Fig. 1). Addressing the instability of optical properties of elastomers in deformation area could provide great opportunities for flexible and wearable optoelectronics.

In this work, full-color persistent luminescence elastomers were prepared by blending polyacrylamide containing organic quaternary phosphonium salts and PVA into a polydimethylsiloxane (PDMS) matrix. The amino group and positively charged quaternary phosphonium salts in RTP polymers, the alcohol group in the PVA, and the ether linkage in PDMS can interact with each other through hydrogen bonding, ion–dipole, and dipole–dipole interactions, thus endowing the prepared elastomer with strong mechanical properties and intense RTP properties. The obtained elastomers exhibit satisfactory mechanical properties, high optical transparency, and bright persistent luminescence. Importantly, their afterglow RTP can be well retained in the large-area deformation region and after multiple rounds stretching. To our knowledge, this work represents an example of multi-color persistent RTP elastomers with robust and stable optical properties. Finally, we demonstrated the complex knitting and anti-counterfeiting labels with high stretchability and stable persistent RTP (Fig. 1).

## Results

### Preparation of persistent RTP elastomers

PDMS was selected as the matrix in this study because it is an elastomer with high flexible elasticity, optical transparency, chemical stability, and excellent resistance to high and low temperature[34]. Organic quaternary phosphonium derivatives were chosen as the phosphorescent monomers because they were reported to exhibit intense persistent RTP from single molecule, and their afterglow colors can be readily tuned by appropriate chemical modification[35–37]. Thus, six monomers of 4-(but-3-en-1-yl-N,N-dimethylaniline)diphenylphosphoranyl bromide (M1), but-3-en-1-yl(4-methoxynaphthalen-1-yl)diphenylphosphonium bromide (M2), but-3-en-1-yl(phenanthren-9-yl)diphenylphosphonium bromide (M3), but-3-en-1-yl(4-fluoronaphthalen-1-yl)diphenylphosphonium bromide (M4), 4-(but-3-en-1-yl-1-naphthonitrile)diphenylphosphonium bromide (M5), but-3-en-1-yldiphenyl(pyren-2-yl)phosphonium bromide (M6) were synthesized according to the previous reported methods[38]. Time-Dependent Density Functional Theory (TDDFT) calculations, using the B3LYP functional and def2-SVP basis set, were conducted to simulate the phosphorescence spectra of monomers M1-M6. This computational approach provided valuable insights into the energy levels of their lowest-lying triplet states (Supplementary Fig. 27). The results revealed distinct triplet energy levels for these monomers, supporting their capacity to generate multi-color RTP.

The copolymerization of M3 and acrylamide was conducted at various molar ratios, including 1:400, 1:200, 1:100, 1:50, and 1:10, resulting in a series of persistent P3 RTP polymers. As illustrated in Supplementary Fig. 28, the phosphorescence peaks for these polymers are almost identical. Upon evaluation of emission decay times, it was observed that the 1:50 molar ratio had the most substantial RTP lifetime of 1090 ms (Supplementary Fig. 28). Consequently, a series of persistent RTP polymers (P1-P6) were prepared by copolymerization of

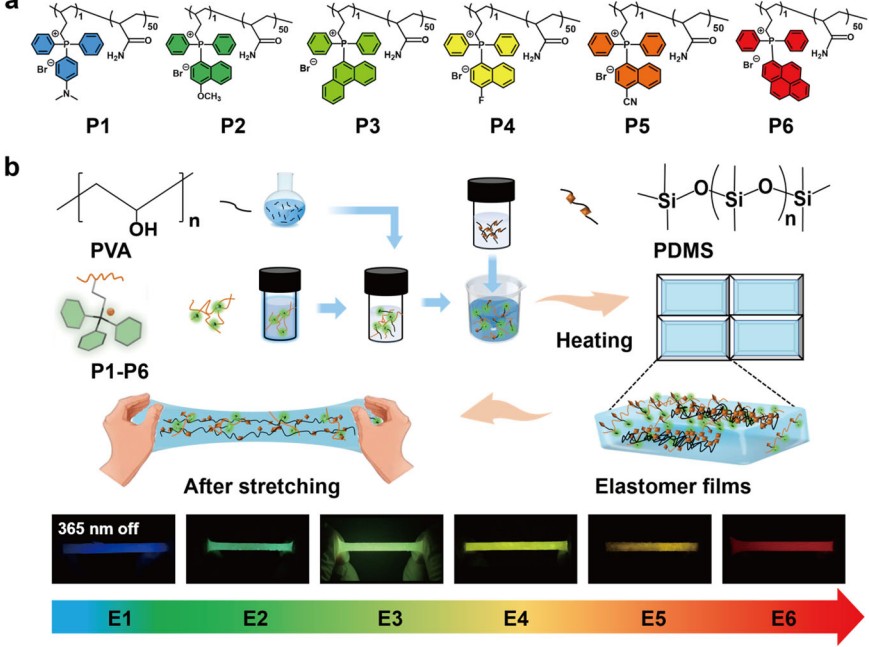

**Fig. 1 | Chemical structures of P1-P6 and the preparation procedure of persistent RTP elastomers. a** Chemical structures of amorphous RTP copolymers P1-P6. **b** Schematic representation for the preparation of full-color persistent room temperature phosphorescent elastomers.

monomers with acrylamide in molar ratio of 1: 50. The detailed synthetic procedures are illustrated in the supplementary information. The obtained monomers were characterized by $^1$H, $^{13}$C NMR spectroscopy, and high-resolution mass spectrometry (ESI-MS). The obtained polymers were characterized by gel permeation chromatography (GPC) and $^1$H NMR (Supplementary Figs. 1–26 and Table S1).

Firstly, we tried to fabricate persistent RTP elastomer by simply mixing persistent RTP polymer with PDMS precursors (Sylguard-184A and Sylguard-184B). It was found that the distribution of RTP polymer in PDMS matrix was inhomogeneous, which is due to the immiscibility between hydrophilic RTP polymer and hydrophobic PDMS. To achieve homogeneous luminescence in the elastomers, the uniform dispersion of RTP polymers in the PDMS matrix is very important. Thus, to overcome this immiscibility barrier, PVA was incorporated into the elastomer because of its ability to mediate binding between the RTP polymer and PDMS through hydrogen bonding and van-der-Waals interactions[39–42]. The mechanism diagram for grafting of PDMS and PVA was depicted in Supplementary Fig. 31. Firstly, different RTP polymers (P1-P6) and PVA were refluxed in the aqueous solution for 6 h. Then, Sylguard-184A was added into the mixture for 2 min stirring, and Sylguard-184B was subsequently added. Next, the mixture was stirred for ten minutes under the 365 nm irradiation. Lastly, the viscous mixture was transferred into a fluorided mold, and elastomer films (E1-E6) were obtained after cooling. The transmittance of the prepared thin films is >85% at the visible light region (410–750 nm), indicating uniform dispersion of RTP polymers in elastomers and their high transparency (Fig. 2a). Such high transparency is beneficial for the afterglow emission generated by RTP polymers to readily penetrate the elastomer matrix. It is noteworthy that increasing the contents of RTP polymers or PVA may influence the photophysical and transmittance properties of the prepared elastomers (Supplementary Figs. 32 and 33).

## Photophysical property

The photophysical properties of P1-P6 were first studied by steady-state and delayed PL spectra. The detailed photophysical data were given in the Supplementary Information (Supplementary Fig. 30 and Supplementary Tab. 2). All the polymers exhibit intense absorption peaks below 500 nm, which are attributed to intramolecular π-π, π-π* or n-π* transitions, as depicted in Supplementary Fig. 30. Upon UV excitation, all polymers showed blue or green fluorescence in the solid state under ambient condition. The delayed PL spectra exhibited that the phosphorescence peaks of these polymers are located at 450 nm, 490 nm, 505 nm, 520 nm, 555 nm, and 620 nm, respectively. Their

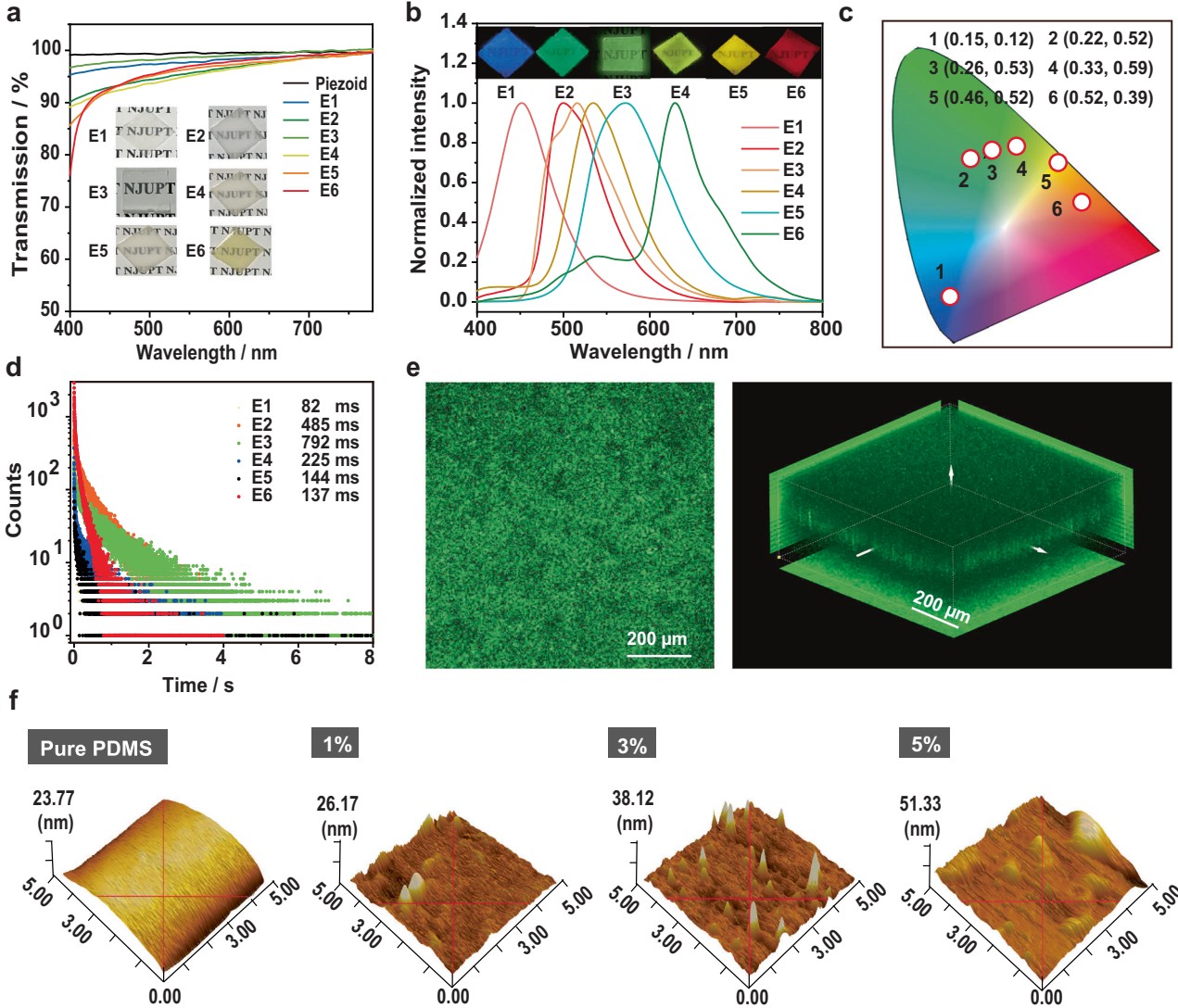

**Fig. 2 | Characterization of RTP elastomers. a** Transmittance spectra of E1-E6. **b** Phosphorescence spectra of E1-E6 ($\lambda_{ex}$ = 365 nm). **c** CIE chromaticity diagram for E1-E6. **d** Lifetime decay curves of E1-E6. **e** Confocal fluorescence images of E3 ($\lambda_{ex}$ = 405 nm). **f** AFM images of PDMS and E3 films with different contents of RTP polymer.

emission decay times were measured to be 135 ms, 679 ms, 1090 ms, 413 ms, 275 ms, and 176 ms, respectively (Supplementary Fig. 30). Thus, different afterglow colors from deep blue to red were observed with naked eyes when the UV source was removed.

To confirm the long-lasting luminescence of these polymers originated from triplet state rather than singlet state, we conducted a detailed analysis of the temperature dependence of phosphorescence in a P3 sample (1/50). As depicted in Supplementary Fig. 29, over the range of 77 to 300 K, a noticeable decrease in emission intensity was observed with an increase in temperature. Further, the lifetime decay curves of P3 decreased from 1276 ms to 759 ms as the temperature increased from 77 K to 300 K. These results derived from temperature-dependent phosphorescence spectra and lifetime decay curves substantiate the conclusion that the long-lasting luminescence originates from the triplet state rather than the singlet state. The intense interactions among polymeric chains through hydrogen bonding constructed a network that can provide a rigid environment to restrict the molecular vibrations, which are responsible for reducing the non-radiative transition and promoting RTP emissions as demonstrated in the previous study[43].

The elastomers E1-E6 exhibited the similar delayed PL spectra and RTP colors with P1-P6, and the Commission International de l'Eclairage (CIE) coordinates for them are (0.15, 0.12), (0.22, 0.52), (0.26, 0.53), (0.33, 0.59), (0.46, 0.52), and (0.52, 0.39), respectively (Fig. 2b, c). Particularly, it is worth noting that the CIE coordinate of delayed PL of E1 is located at deep blue region, which is very difficult to achieve for organic persistent RTP. The emission lifetimes were reduced to 82 ms, 485 ms, 792 ms, 225 ms, 144 ms, and 137 ms, because molecular interactions among polymeric chains of monomers were weakened after they were blended into PDMS matrix (Fig. 2d). Among them, E3 exhibited the brightest emission and longest RTP lifetime (Supplementary Movie 1 and 2), thus it was studied in detail as a model in the following sections.

To confirm the homogeneous afterglow RTP in the elastomers, confocal fluorescence microscopy was used to demonstrate their uniform distribution. As shown in Fig. 2e and Supplementary Fig. 18, we can see the homogeneous emission in the confocal fluorescence images, indicating that RTP polymers were uniformly dispersed in PDMS (Supplementary Fig. 34). Z-scans luminescence imaging of E3 further confirms the high uniformity as well. Furthermore, scanning electron microscopy (SEM) and energy-dispersive X-ray spectroscopy (EDS) were employed to analyze the brittle fracture section of E3. As indicated by the EDS mapping image, phosphorous (P) elements, which are solely derived from the P3 polymer, are uniformly distributed within the PDMS matrix (Supplementary Fig. 35). This evidences the homogeneous dispersion of the RTP polymer within the PDMS matrix. In addition, atomic force microscope (AFM) was used to detect the surface morphology of E3. Figure 2f showed that pure PDMS film has a very smooth surface, while E3 exhibited a roughness surface. However, no obvious accumulation of RTP polymer aggregates can be detected. The hydrogen bonding, ion–dipole, and dipole–dipole interactions among amino group and positively charged quaternary phosphonium salts in the RTP polymers, alcohol group in the PVA, and ether linkage in PDMS made the RTP polymers distribute uniformly in the PDMS and prevented them from aggregating[39].

## Mechanical properties

The tensile tests were carried out to study the mechanical properties of these persistent RTP elastomers. First, the tensile stress–strain curves of E3 with different thicknesses from 0.5 mm to 2.0 mm were measured, and the results show that the elastomer with 1.0 mm thickness exhibited the best performance among them, which can be stretched to 153% of its original length (Fig. 3a). Therefore, the elastomers of E1-E6 were prepared with 1.0 mm thickness for mechanical property investigation. In addition, considering that the mechanical properties

are normally sensitive to the strain rates, thus different strain rates were applied for the study, and the results were recorded as shown in Fig. 3b. We can see that the engineering stress of a 1.0 mm thickness E3 decreased when the strain rate was decreased, and the strain to failure increased with a decrease in strain rate. Hence, the strain rate of 20 mm min$^{-1}$ was used for the comparison of mechanical properties of E1-E6. As exhibited in Fig. 3c, these persistent RTP elastomers E1-E6 show better performance in stretchability (116–173%) than that of pure PDMS film (95%). This might be because the addition of ionic RTP polymers and PVA can provide extra hydrogen bonding, ion–dipole, and dipole–dipole interactions in the PDMS matrix, thus resulting in the increase of stretchability.

Next, the stability of optical property was studied by stretching elastomers with large strain. The mechanical robustness of the persistent RTP elastomers were investigated by the tensile stretch-release cycles (Supplementary Movie 3). Figure 3d showed that no remarkable degradation of the tensile profiles was observed for E3 after 100 cycles of 25% strain stretch-release cycles. Simultaneously, it is exciting to see that only 15.4% and 6.5% degradation in phosphorescence intensity and lifetime of E3 were detected before and after 100 cycles of 25% strain stretch-release cycles (Fig. 3e, f). Therefore, we can conclude that these persistent RTP elastomers are mechanically robust and mechanical deformation only has little effect on their phosphorescence properties.

As shown in Fig. 4a, the strong persistent RTP can be clearly observed when the elastomer films were stretched, even in the deformation region. Thus, the detailed investigation on the photophysical properties of E3 in the deformation region was studied. As shown in Fig. 4b, the photophysical properties of E3 in the different stretched states (from 1 to 78%) were investigated. The results showed that the RTP intensity and emission lifetimes decreased to 92 and 84% of its original value, suggesting that their optical properties have good resistance against mechanical deformations (Supplementary Fig. 36). Impressively, the RTP and mechanical properties of E3 can be well retained even under water (Fig. 4c). These fantastic features make these persistent RTP elastomer films of E1-E6 perfect candidates for practical usage (Supplementary Movie 4). It is known that the generation of RTP in these materials is primarily due to the fact the polyacrylamide can effectively immobilize the phosphors due to the presence of hydrogen bond cross-linking between the polymeric chains[44]. When these films undergo stretching, the strain imposed may sever the hydrogen bond cross-links amongst polymeric chains in some extent. This disruption weakens the immobilization of phosphors, thereby leading to a reduction in RTP intensity and emission lifetimes.

## Potential applications

The elastomers with unique persistent RTP properties and good mechanical properties can be applied in various fields, such as display, fashion, or anti-counterfeiting. The elastomers of E1, E3, and E6 were prepared as persistent RTP lines, which are almost transparent under the daylight, but showing strong RTP emissions after ceasing 365 nm UV light (Supplementary Fig. 37). As shown in Fig. 5a, the persistent RTP of these lines can be well maintained after multiple times bending and twisting, indicating that they can be directly used as functional lines to fabricate different complicated knitting. As shown in Fig. 5b, a complicated knot with blue, green, and red afterglow colors was fabricated for fashion and aesthetic purposes.

In addition, the mixture of RTP polymers, PVA, and PDMS in a certain proportion can be used as security inks to print stretchable anti-counterfeiting labels through high-resolution ink-jet printing system. As shown in Fig. 5c, a multicolor label of lotus flower with a size of 5 cm × 5 cm was printed on a PDMS substrate. In Fig. 5d, bending and stretching the badge has almost no effect on its persistent RTP properties, which shows better performance to previous anti-counterfeiting labels[45,46]. Such robust optical properties of

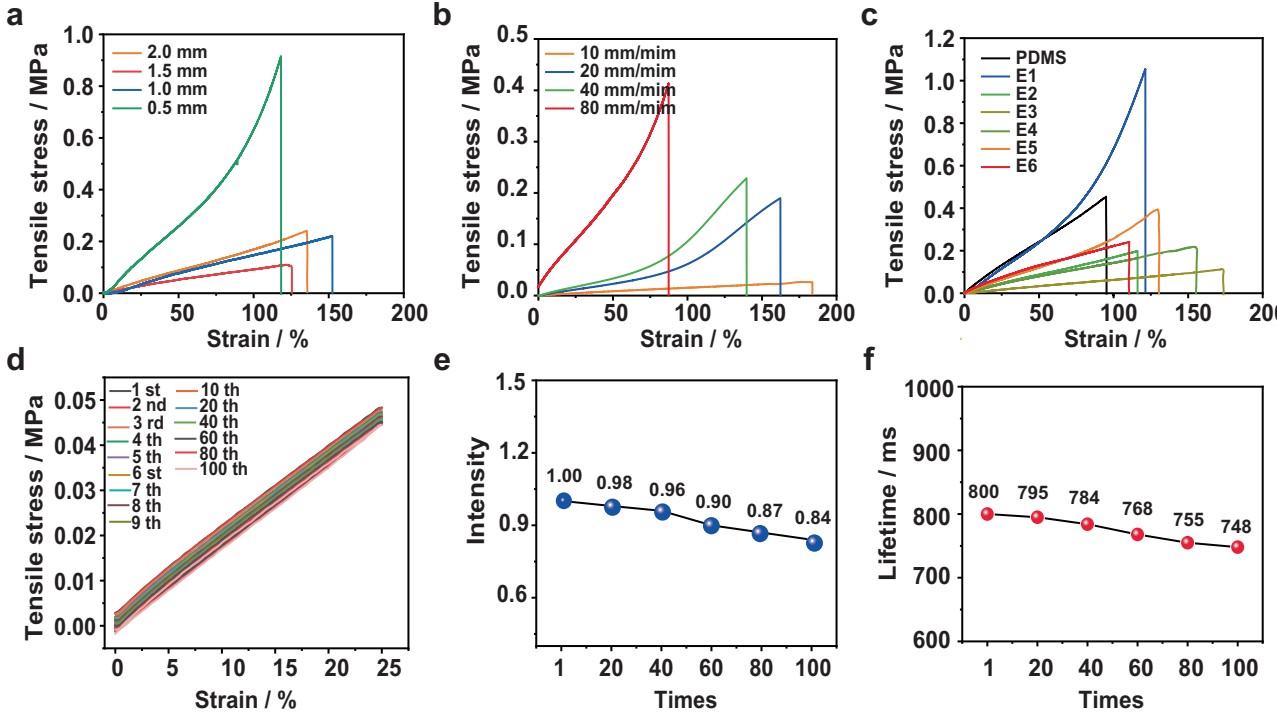

**Fig. 3 | Mechanical properties of RTP elastomers. a** Stress-strain curves of E3 with different thickness (0.5–2.0 mm). **b** Stress-strain curves for E3 (1.0 mm) samples when stretched at different rates. **c** Stress-strain curves of various E1-E6 elastomers (1.0 mm). **d** Cyclic loading and recovery of E3 (1.0 mm). **e** Changes of E3 (1.0 mm) in phosphorescence intensity after multiple-times stretching. **f** Changes of E3 (1.0 mm) in emission lifetime after multiple-times stretching.

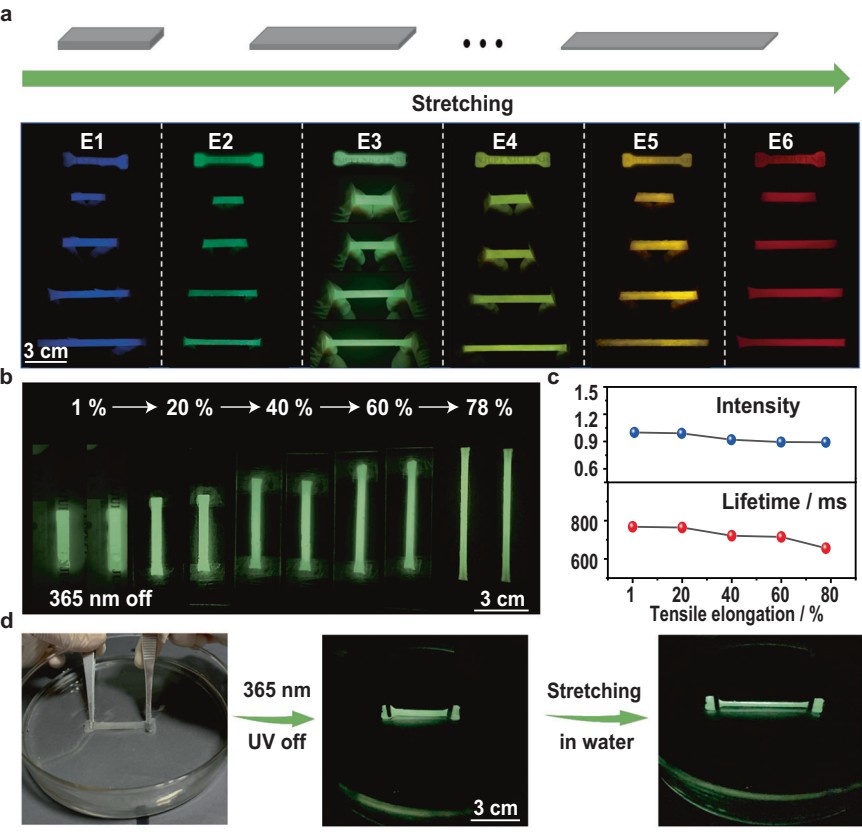

**Fig. 4 | RTP performance of E1-E6 under strain. a** Schematic illustration of stretching E1-E6 and their RTP photographs. **b** RTP afterglow emission of E3 under strain. **c** Changes of E3 in phosphorescence intensity and emission lifetime under strain. **d** RTP afterglow photograph of stretched E3 under water.

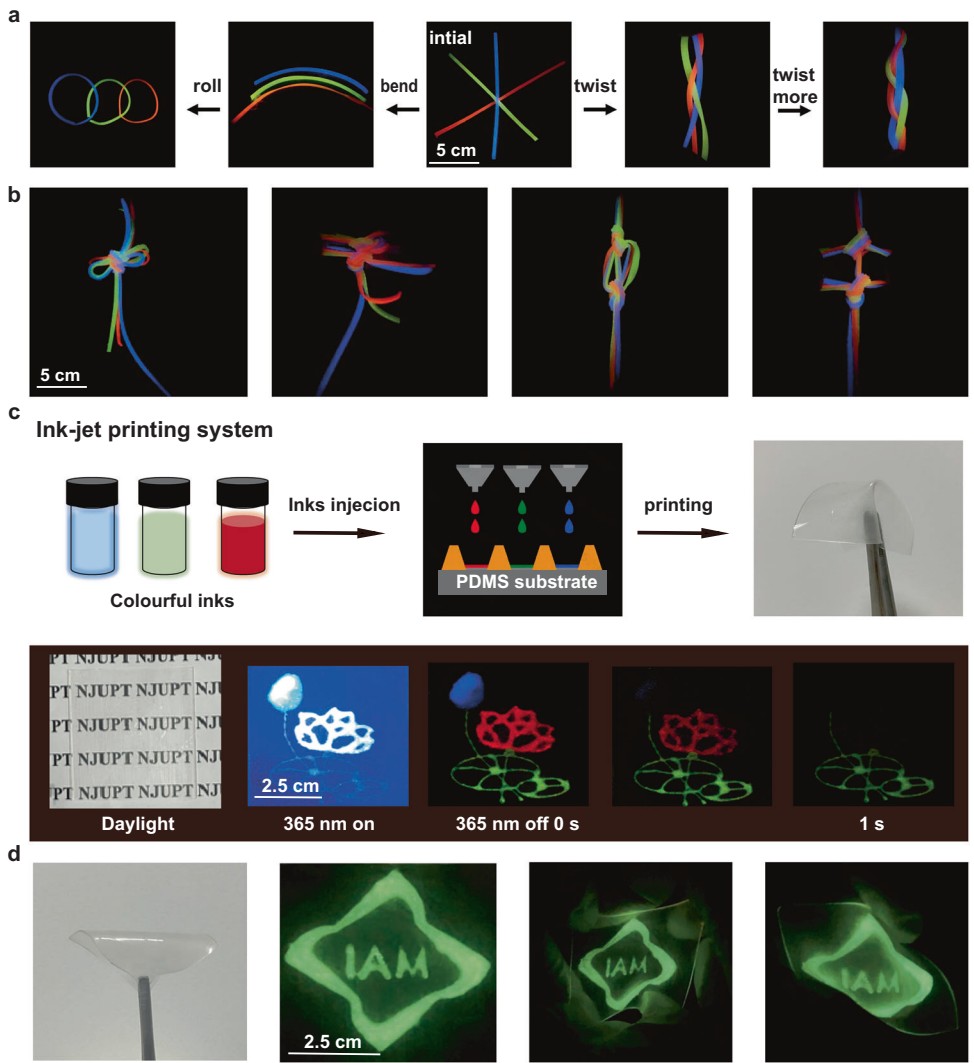

**Fig. 5 | Potential applications of RTP elastomers. a** Photographs of bending, rolling, and twisting E3 after removing 365 nm excitation. **b** Colorful afterglow images of different fabrics. **c** Schematic illustration the microelectronic printing and photographs of printed labels on PDMS substrate by using E1, E3, and E6. **d** Photographs of anti-counterfeiting labels under different stretched states.

these persistent RTP elastomers against various mechanical deformation indicate their great potential in real-world applications.

## Discussion

In summary, we present a concise and facile strategy to fabricate persistent RTP elastomers with high stretchability and robust optical property by blending ionic RTP polymers and PVA into PDMS matrix. Multicolor persistent RTP elastomers can be easily achieved by modifying the chemical structure of monomers to display different phosphorescence colors from blue to red. In addition, the prepared elastomers exhibit good mechanical properties compared to pure PDMS film, attributed to the addition of ionic RTP polymers and PVA, which provide extra hydrogen bonding, ion–dipole, and dipole–dipole interactions in the PDMS matrix. Importantly, their optical properties can be well maintained even after multiple cycles of bending, twisting, and stretching. Based on these fantastic features, we successfully prepared complex knots with different afterglow colors, demonstrating their potential in flexible display and fashion design. Moreover, the persistent RTP elastomers were used to print an anti-counterfeiting label on the PDMS substrate, which can retain intense afterglow emission upon bending and stretching. It is believed that the development of persistent luminescence elastomers with robust and stable optical properties will greatly broaden the range of organic RTP materials and promote their practical applications in wearable optoelectronic devices, flexible displays, and advanced anti-counterfeiting areas.

## Methods

### Measurements

$^1$H NMR (400 MHz) and $^{13}$C NMR (100 MHz) spectra were recorded on a Bruker ACF400 spectrometer at 298 K using deuterated solvents (CDCl$_3$ or DMSO-$d_6$). The UV-visible absorption spectra were measured by Shimadzu UV-2600 UV-vis spectrophotometer. The steady-state fluorescence and phosphorescence spectra were measured using Hitachi F-4700. The lifetimes were obtained on an Edinburgh FLS980 fluorescence spectrophotometer equipped with a Xenon arc lamp (Xe900) and a microsecond flash-lamp (uF900). The powder X-ray diffraction patterns were collected by D8 Advanced (Bruker) using Cu-Kα radiation. Aqueous gel permeation chromatography (GPC) was performed on a Series Samples used 0.05 mol/L NaNO$_3$/H$_2$O as mobile phase at 1.0 mL min$^{-1}$ flow rate. The patterns were produced by a microelectronic printer (Shanghai Mi Fang Electronic Technology Co., Ltd). PVA-1799 was used as the calibration standard. The crystalline samples were obtained from slow evaporation of a mixture of hexane/dichloromethane (1:1, v/v).

## Preparation of transparent elastomer films (E1-E6)

Firstly, The RTP polymer P1-P6 (40 mg, 30 mg, 30 mg, 40 mg, 60 mg, 60 mg, respectively.) were dissolved in 2 mL of deionized water. Subsequently, PVA (10 mg) was refluxed in the aqueous solution for 6 h. Then, Sylguard-184A (1000 mg) and Sylguard-184B (100 mg) were added into the mixture. The PDMS-PVA-P3 mixture was stirred for ten minutes under the 365 nm irradiation. Lastly, the viscous mixture was transferred into a fluoridated mold, heating slowly from 50 °C to 120 °C on the heat table for 12 h, and the elastomer films (E1-E6) were obtained after cooling.

## Preparation of RTP lines

Firstly, P3 (30 mg) and PVA (10 mg) were dissolved in 2 mL of deionized water. Subsequently, the Sylguard-184A (1000 mg) and Sylguard-184B (100 mg) were poured into the mixture. The PDMS-PVA-P3 mixture was stirred for ten minutes under the 365 nm irradiation. Lastly, the viscous mixture was transferred into a line fluoridated mold, heating slowly from 50 °C to 120 °C on the heat table for 12 h, and the RTP lines were obtained after cooling.

## Data availability

The datasets generated during and/or analyzed during the current study are available from the corresponding author on request. Source data are provided with this paper.

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

## Acknowledgements

We gratefully acknowledge the financial support from National Funds for Distinguished Young Scientists (61825503), National Natural Science Foundation of China (62288102, 62322508, and 62075101), National Key R&D Program of China (2022YFA1204404), and Natural Science Foundation of Jiangsu Province of China (BK20200095).

## Author contributions

Y.M., W.H., and Q.Z. conceived the idea for this work and designed the experiments. J.W. contributed to the synthesis work, photophysical property study, measurements, and application study. M.Z., J.L., T.D., P.D., C.L., J.D., X.Z., S.Z., L.G., and H.W. contributed to implementation of the experiments. S.J.L. revised the manuscript and provided suggestions. All authors discussed the results and commented on the manuscript at all stages.

## Competing interests

The authors declare no competing interests.
