## [Peer Review File · Nature Communications]

Full-color persistent room temperature phosphorescent elastomers with robust optical propertiesREVIEWER COMMENTS

Reviewer #1 (Remarks to the Author):

In the manuscript, authors describe the preparation and properties of RTP elastomers with PDMS as matrix and the mixture of Px/PVA as dispersion phases, and RTP emission is from dispersed PX particles. Where PDMS is only a dispersion matrix and no significant interactions with the triplet chromophores because the dispersion of Px is not at the molecular level, otherwise, the chain segment motions seriously impair the thermal stability of triplet chromophores and greatly quench RTP. In fact, the loose networked polymers act as poorer stabilizing effect than orientated polymers. Since the very poor compatibility between PDMS and dispersion particles, the film mechanical properties are rather inferior so that the very low break elongation and strength are observed although the low extension rate like plastics measurement is used. In reference 39, the use of stearic acid is as a surfactant of PVA to improve the interface miscibility with PDMS, if it could be verified that there is some grafting between PVA and PDMS under high temperature, the improvement effect of PVA on the miscibility of Px and PDMS is rational. On the other hand, in reference 32 and Yang' another reference (Chemical Engineering Journal 447 (2022) 137458), the elastic RTP samples are demonstrated at high speed cyclic deformation, and no RTP change under constant elongation deformation.

Nevertheless, this work is a beneficial attempt in elastic RTP polymers and is also enlightening, especially their abundant RTP colors. Therefore, I recommend its publication in Nat. Commun. after revisions.

1. In the abstract, "good mechanical properties" should be modified as "unique" or "elastic" to correspond to following "flexible". The meaning of "good" is very different for plastics, rubbers, and fibers, etc.

2. Line 53-57, suggesting the revision as "RTP polymers with good flexibility, large-area processability, and producibility were developed by incorporating small-molecule phosphors into a commercially available polymer matrix."

3. Line 81, "This is because the long flexible polymer chains of RTP copolymers can resist external stretch to some extent" where flexible should be "rigid" because polyacrylamide is not flexible chain in aggregate state. Flexible and resist is contradictory.

4. Line 91, "The obtained elastomers exhibit excellent mechanical properties," neither elongation and strength are poor, the word "excellent" is not proper at least.

5. Line 94. To our knowledge, this work represents the first example of persistent RTP elastomer with robust and stable optical properties. Revising "the first" as a new or "robust and stable optical" as multi-color RTP.

6. line 100, PDMS was selected as the matrix in this study because it is the most commonly used elastomer due to its elasticity, optical transparency, and chemical stability. PDMS is special and is not common elastomer. Alternatively, --- because it is an elastomer with high flexible elasticity, optical

transparency, chemical stability and excellent resistance to high and low temperature.

7. Line 112, prepared by copolymerization of monomers with acrylamide in molar ratio of 1 : 50. Where is the monomer ratio equal to the unit ratio in the co-polymer? I think vinyl monomers containing the salts is less co-polymerized and the actual content is not clearly described.

8. Line 298, Preparation of RTP fibers. Firstly, P3 (30 g) and PVA (10 mg) were dissolved in 2 mL of deionized water. Subsequently, the Sylguard-184A (1000 mg) and Sylguard-184B (100 mg) were poured into. Lastly, the viscous mixture was transferred into a line fluoridated mold, heating slowly from 50 °C to 120 °C on the heat table for 12h, then the RTP fibers were obtained.

P3 (30 g), is this amount correct? Also, transferred into a line fluoridated mold, and is the obtained material fiber? According to the label in Figure 5a, the diameter is millimeter size at least. If naming fiber as the large aspect ratio, the extruded rod and tube become fibers. Fibers should at least be characterized by orientation and anisotropy.

9. Line 170, Confocal fluorescence and Z-scan luminescence images only reflect the light-emitting uniformity from the matrix, which does not represent the dispersion uniformity of RTP polymers (dopants) in any matrix (including PDMS). Light diffuses and overlaps in all directions, and Px particles (phases) are heterogeneously formed and dispersed in PDMS. I think SEM on brittle fracture section is suitable. The visual uniform light emission does not need very uniform phase dispersion.

Reviewer #2 (Remarks to the Author):

The manuscript described a series of room temperature phosphorescent elastomers by blending ionic RTP polymers and polyvinyl alcohol into polydimethylsiloxane (PDMS) matrix. And they were used as functional fibers to fabricate different complicated knitting and applied in anti-counterfeiting. It is a concise and facile strategy to fabricate persistent RTP elastomers, and the application in functional fibers is new and interesting. However, the manuscript may be not suitable enough for publication in the journal of Nature communications considering its innovations and workload.

1. The design strategy of the six monomers of M1-M6 should be further declared. And the principle of the luminescent changes among P1-P6 should be explained.
2. The author represented the decrease of RTP intensity and emission lifetimes between original films and stretching films, but the reason was not explained.
3. The monomers utilized in this work have been reported and the mechanical properties among RTP elastomers have not shown further innovations in either luminescent properties or stretchable materials. The language expression utilized should also be further modified. Therefore, I think this work does not meet the standards required in Nature Communication.

4. There are many problems in the English expression of the article such as tenses, singular and plural of nouns and so on, so it is recommended to revise it carefully. And here are some errors.

4.1 In the part of "Abstract", "Persistent room temperature phosphorescence (RTP) materials with good mechanical properties and robust optical properties have great potential in flexible electronics and photonics," should be corrected to be "Persistent room temperature phosphorescent (RTP) materials with good mechanical properties and robust optical properties have great potential in flexible electronics and photonics,".

4.2 In the second paragraph of "Introduction", "The constant mechanical deformations in practical applications make those materials cannot meet the requirements for real-world usage." should be corrected to be "The constant mechanical deformations in practical applications make those materials dissatisfy the requirements for real-world usage."

4.3 In the third paragraph of "Introduction", "Alternatively, by blending amorphous RTP copolymers into elastomer matrices might be an effective way to obtain stretchable films with stable persistent luminescence properties." should be corrected to be "Alternatively, blending amorphous RTP copolymers into elastomer matrices might be an effective way to obtain stretchable films with stable persistent luminescence properties."

Reviewer #3 (Remarks to the Author):

In this manuscript, the authors report a series of stretchable and multicolor persistent RTP elastomers, they exhibited high optical transparency under daylight and bright persistent luminescence after the removal of 365 nm excitation. Even after large-area and multiple times mechanical deformations, the optical properties can be well retained, which is previously unprecedented. This is an interesting work, the demonstrations are well designed, and the results are solid. It can provide important information for developing RTP elastomers, and it will be of interest to a wide readership. Based on the rational material preparation and the outstanding optical and mechanical performance, it can be recommended for publication in Nature Communications after minor revision.

1. In this work, only ¹H NMR and ¹³C NMR were used to characterize the monomers. HRMS analysis of each of monomer is recommended to be used for further demonstration of their chemical structures.

2. UV-visible absorption spectra of P1-P6 are not provided.

3. Why 1/50 was chosen for polymerization?

4. How does the authors confirm the long-lasting luminescence of P1-P6 from triplet state rather singlet state?

5. How the quantum yields were determined in this work?

6. In Supplementary Tab.3, I think the P1-P6 should be E1-E6?

Point-by-point response to the reviewers:

Reviewer #1 (Remarks to the Author):

In the manuscript, authors describe the preparation and properties of RTP elastomers with PDMS as matrix and the mixture of Px/PVA as dispersion phases, and RTP emission is from dispersed PX particles. Where PDMS is only a dispersion matrix and no significant interactions with the triplet chromophores because the dispersion of Px is not at the molecular level, otherwise, the chain segment motions seriously impair the thermal stability of triplet chromophores and greatly quench RTP. In fact, the loose networked polymers act as poorer stabilizing effect than orientated polymers. Since the very poor compatibility between PDMS and dispersion particles, the film mechanical properties are rather inferior so that the very low break elongation and strength are observed although the low extension rate like plastics measurement is used. In reference 39, the use of stearic acid is as a surfactant of PVA to improve the interface miscibility with PDMS, if it could be verified that there is some grafting between PVA and PDMS under high temperature, the improvement effect of PVA on the miscibility of Px and PDMS is rational. On the other hand, in reference 32 and Yang' another reference (Chemical Engineering Journal 447 (2022) 137458), the elastic RTP samples are demonstrated at high-speed cyclic deformation, and no RTP change under constant elongation deformation. Nevertheless, this work is a beneficial attempt in elastic RTP polymers and is also enlightening, especially their abundant RTP colors. Therefore, I recommend it publication in Nat. Commun. after revisions.

Response: We would like to thank the precious time devoted to the reviewing process of the reviewer. We appreciate the reviewer's constructive comments and suggestions. Previous literatures (*J. Polym. Sci. A Polym. Chem.*, **2009**, 47, 5272; *J. Appl. Polym. Sci.*, **2002**, 85, 957) have successfully demonstrated the high-temperature grafting process between PVA and PDMS. The grafting of sufficient volumes of PDMS branches onto PVA results in continuous hydrophilic PVA/PDMS domains, enhancing the uniformity of Px dispersion into the PDMS matrix.

In addition, it is observed that the deformation can lead to RTP quenching in the referenced work (*Adv. Sci.*, **2022**, 9, 2103402). However, it is worth noting that in our

work, we have demonstrated that the elastomers in this work can maintain their persistent RTP properties under large mechanical strain, which is a significant advancement compared to previous works. Regarding another referenced study (*Chem. Eng. J.*, **2022**, 447,137458), they did not explore changes in RTP under high-speed cyclic or constant elongation deformation. Lastly, we are pleased to note that the reviewer appreciates the array of persistent RTP colors that we were able to achieve in this study. We believe these findings will be valuable for future research in the field. We hope that our work will meet the high standards of *Nature Communications* after careful revisions.

1 In the abstract, “good mechanical properties” should be modified as “unique” or “elastic” to correspond to following “flexible”. The meaning of “good” is very different for plastics, rubbers, and fibers, etc.

Response: We appreciate the reviewer’s comment. According to the suggestion, “good mechanical properties” has been revised to “unique mechanical properties” in the revised manuscript.

2 Line 53-57, suggesting the revision as “RTP polymers with good flexibility, large-area processability, and producibility were developed by incorporating small-molecule phosphors into a commercially available polymer matrix.”

Response: We appreciate the reviewer’s comments. According to the suggestion, the statement in lines 53-57 has been revised to “RTP polymers with good flexibility, large-area processability, and producibility were developed by incorporating small-molecule phosphors into a commercially available polymer matrix” in the revised manuscript.

3 Line 81, “This is because the long flexible polymer chains of RTP copolymers can resist external stretch to some extent” where flexible should be “rigid” because polyacrylamide is not flexible chain in aggregate state. Flexible and resist is contradictory.

Response: We appreciate the reviewer's comments. We agree that the term "flexible" may lead to confusion in this context. We intended to imply that the long polymer chains of RTP copolymers can adapt to external forces due to their inherent molecular flexibility, which contributes to their ability to resist deformation. However, considering the reviewer's point, it would be more precise to delete the term "flexible". Hence, the statement in line 81 has been revised to "This is because the long polymer chains of RTP copolymers can resist external stretch to some extent." We hope this revision eliminates any potential misunderstanding.

4 Line 91, The obtained elastomers exhibit excellent mechanical properties," neither elongation and strength are poor; the word "excellent" is not proper at least.

Response: We appreciate the reviewer's comments. According to the suggestion, we have revised "excellent" to "satisfactory" in line 91 in the revised manuscript.

5 Line 94. to our knowledge, this work represents the first example of persistent RTP elastomer with robust and stable optical properties. Revising "the first" as a new or "robust and stable optical" as multi-color RTP.

Response: We appreciate the reviewer's comments. According to the suggestion, we have revised "the first" to "a new" and "multi-color" has been added in line 94 in the revised manuscript.

6 line 100, PDMS was selected as the matrix in this study because it is the most commonly used elastomer due to its elasticity, optical transparency, and chemical stability. PDMS is special and is not common elastomer. Alternatively, --- because it is an elastomer with high flexible elasticity, optical transparency, chemical stability and excellent resistance to high and low temperature.

Response: We appreciate the reviewer's comments. According to the suggestion, the statement in line 100 have been revised to "PDMS was selected as the matrix in this study because it is an elastomer with high flexible elasticity, optical transparency,

chemical stability and excellent resistance to high and low temperature” in the revised manuscript.

7 Line 112, prepared by copolymerization of monomers with acrylamide in molar ratio of 1: 50. Where is the monomer ratio equal to the unit ratio in the co-polymer? I think vinyl monomers containing the salts is less co-polymerized and the actual content is not clearly described.

Response: We appreciate the reviewer’s comments. The 1:50 ratio mentioned in the manuscript refers to the feed ratio used for copolymerization. The actual copolymerization ratio can be determined through the analysis of ^1H NMR spectra. Figures R1-R6 demonstrate that the actual copolymerization ratios of P1-P6 are determined to be 1:61, 1:57, 1:55, 1:63, 1:67, and 1:56, respectively. These values have been added in the revised supplementary manuscript.

Figure R1 ^1H NMR spectrum of P1 in D_2O .

Figure R2 ^1H NMR spectrum of P2 in D_2O .

Figure R3 ^1H NMR spectrum of P3 in D_2O .

Figure R4 ^1H NMR spectrum of P4 in D_2O .

Figure R5 ^1H NMR spectrum of P5 in D_2O .

Figure R6 ^1H NMR spectrum of P6 in D_2O .

8 Line 298, Preparation of RTP fibers. Firstly, P3 (30 g) and PVA (10 mg) were dissolved in 2 mL of deionized water. Subsequently, the Sylguard-184A (1000 mg) and Sylguard-184B (100 mg) were poured into. Lastly, the viscous mixture was transferred into a line fluoridated mold, heating slowly from 50 °C to 120 °C on the heat table for 12h, then the RTP fibers were obtained.

P3 (30 g), is this amount correct? Also, transferred into a line fluoridated mold, and is the obtained material fiber? According to the label in Figure 5a, the diameter is millimeter size at least. If naming fiber as the large aspect ratio, the extruded rod and tube become fibers. Fibers should at least be characterized by orientation and anisotropy.

Response: We appreciate the reviewer’s attention to detail and for pointing out the typographical error in the manuscript. The correct amount of P3 should indeed be 30 mg, not 30 g. We apologize the oversight and have corrected this in the revised manuscript.

As for your concern regarding the term “fiber”, we agree that the term traditionally refers to materials with high aspect ratios and typically also implies a certain degree of orientation and anisotropy. The term was used in our manuscript to refer to the large

aspect ratio form of the material. However, we understand that this might lead to confusion and we have therefore revised the terminology used in the manuscript. Instead of “fiber”, we now refer to the material as “line” in the revised manuscript.

9 Line 170, Confocal fluorescence and Z-scan luminescence images only reflect the light-emitting uniformity from the matrix, which does not represent the dispersion uniformity of RTP polymers (dopants) in any matrix (including PDMS). Light diffuses and overlaps in all directions, and Px particles (phases) are heterogeneously formed and dispersed in PDMS. I think SEM on brittle fracture section is suitable. The visual uniform light emission does not need very uniform phase dispersion.

Response: We appreciate the reviewer’s comments. After careful consideration, we agree that confocal fluorescence images alone do not suffice to conclusively demonstrate the dispersion uniformity of RTP polymers in PDMS. According to the suggestion, we have carried out scanning electron microscopy (SEM) and energy-dispersive X-ray spectroscopy (EDS) mapping experiments on brittle fracture section to provide more direct evidence of the dispersion uniformity of RTP polymers in the matrix. As indicated by the EDS mapping image, phosphorous (P) elements, which are solely derived from the P3 polymer, are uniformly distributed within the PDMS matrix (Figure R7). This evidences the homogeneous dispersion of the RTP polymer within the PDMS matrix.

These additional data have now been included in the revised manuscript.

Figure R7. a, b SEM image of E3 fracture section. **c** EDS image of E3.

Reviewer #2 (Remarks to the Author):

The manuscript described a series of room temperature phosphorescent elastomers by blending ionic RTP polymers and polyvinyl alcohol into polydimethylsiloxane (PDMS) matrix. And they were used as functional fibers to fabricate different complicated knitting and applied in anti-counterfeiting. It is a concise and facile strategy to fabricate persistent RTP elastomers, and the application in functional fibers is new and interesting. However, the manuscript may be not suitable enough for publication in the journal of Nature communications considering its innovations and workload.

Response: We would like to thank the precious time devoted to the reviewing process of the reviewer. We appreciate the referee's recognition of novelty and interest of this work, particularly in the fabrication of persistent RTP elastomers and their application in functional fibers. Regarding the concerns about the innovation and workload in this work, we would like to provide further clarification.

The key innovations of our work can be summarized in following points:

1. This study outlines a straightforward, highly effective, and universally applicable strategy for the fabrication of RTP elastomers. The method involves mixing ionic RTP polymers and PVA with a PDMS matrix to produce durable RTP elastomers with robust and stable optical properties. Furthermore, through rational molecular design of monomers of organic quaternary phosphonium derivatives, we prepared phosphorescent monomers that can achieve various afterglow luminescence from blue to red. This versatile approach not only simplifies the preparation process of RTP elastomers but also offers a customizable platform for designing materials with tailor-made properties, opening up new possibilities for their application.

2. Compared to previous reports, the persistent RTP elastomers in this work can retain their ultralong RTP even in stretched states. This breakthrough represents a substantial progression in the field, paving the way for an extensive range of RTP material in various optoelectronic areas.

3. We expanded the practical applications of these persistent RTP elastomers by demonstrating their potential in optical anti-counterfeiting. We successfully fabricated complex knitting patterns and anti-counterfeiting labels with high stretchability and

stable persistent RTP properties. This pioneering work showcases the immense potential of these RTP elastomers in wearable optoelectronics, flexible displays, and advanced security applications.

Regarding the workload, we would like to provide further clarification as follows:

1. A substantial portion of our effort was dedicated to ensuring that the RTP polymers were uniformly dispersed within the PDMS matrix, while also maintaining their pronounced RTP properties. This process involved a significant amount of work with various polymer backbones, such as acrylamide, and different intermediates like PVA. Achieving the right balance required rigorous experimentation and optimization.

2. The second major aspect of our work revolved around the design and synthesis of a series of triphenylphosphine derivatives with different triplet energy levels. From this, phosphorescent monomers that cover a full spectrum of colors across the visible light region were selected. Each monomer was then converted into a polymer and subsequently incorporated into a persistent RTP elastomer. This step required careful planning and meticulous synthesis, followed by comprehensive measurements to verify the luminescent properties of each resulting elastomer.

3. Lastly, we devoted significant time and resources to thoroughly characterize the optical and mechanical properties of the persistent RTP elastomers. We further extended our work by exploring the practical applications of these materials, particularly in the field of optical anti-counterfeiting. This involved creating complex knitting patterns and anti-counterfeiting labels using these materials, which required careful design, fabrication, and testing to ensure the stability of the persistent RTP under conditions of deformation.

Overall, while the methods employed may seem straightforward, they entailed substantial effort and extensive refinement to ensure optimal results.

1. The design strategy of the six monomers of M1-M6 should be further declared. And the principle of the luminescent changes among P1-P6 should be explained.

Response: We appreciate the reviewer's comments. The design of monomers M1-M6 was based on our intention to explore the changes in the photophysical properties of the corresponding polymers resulting from the variation of the monomeric units. Each monomer was designed with a different structure to induce diverse triplet excited states. Time-dependent density functional theory (TDDFT) calculations, using the B3LYP functional and def2-SVP basis set, were conducted to simulate the phosphorescence spectra of monomers M1-M6. This computational approach provided valuable insights into the energy levels of their lowest-lying triplet states (refer to Supplementary Fig. 27). The results revealed distinct triplet energy levels for these monomers, supporting their capacity to generate multi-color RTP.

Figure R8. The frontier molecular orbitals of the lowest excited triplet states for various monomers M1-M6, as calculated using the TDDFT with B3LYP functional and def2-SVP basis set.

2. The author represented the decrease of RTP intensity and emission lifetimes between original films and stretching films, but the reason was not explained.

Response: We appreciate the reviewer's comments. The generation of RTP in these materials is primarily due to the fact that the polyacrylamide can effectively immobilize the phosphors due to the presence of hydrogen bond cross-linking between the polymeric chains. When these films undergo stretching, the strain imposed may sever the hydrogen bond cross-links amongst polymeric chains in some extent. This disruption weakens the immobilization of phosphors, thereby leading to a reduction in RTP intensity and emission lifetimes. We hope this explanation provides a clearer

understanding of the observed phenomena and the discussion has been added in the revised manuscript.

3. The monomers utilized in this work have been reported and the mechanical properties among RTP elastomers have not shown further innovations in either luminescent properties or stretchable materials. The language expression utilized should also be further modified. Therefore, I think this work does not meet the standards required in Nature Communication.

Response: We appreciate the reviewer's comments. Although the monomers employed in this work have been previously studied, their application to the development of RTP elastomers is novel and innovation. Our study notably involves blending these ionic RTP polymers and PVA into a PDMS matrix, a method not previously reported. Our study offers an in-depth investigation of the photophysical properties of elastomers under various stretched states, a study that to our knowledge has not been previously reported. Our findings highlight that these RTP elastomers maintain remarkable RTP performance even under high mechanical strain, which is a significant contribution to the RTP materials. We believe that this work represents a breakthrough in the development of high-performance persistent RTP elastomers and is warranted for publication in Nature Communications. In addition, the revised manuscript has been reviewed and improved by a native English speaker for clarity and readability.

4. There are many problems in the English expression of the article such as tenses, singular and plural of nouns and so on, so it is recommended to revise it carefully. And here are some errors.

Response: We appreciate the reviewer's comments. We have thoroughly reviewed and revised the manuscript with the help of a native English speaker who is well versed in scientific writing. We have paid particular attention to the correct usage of tenses, and the singular and plural forms of nouns.

4.1 In the part of “Abstract”, “Persistent room temperature phosphorescence (RTP) materials with good mechanical properties and robust optical properties have great potential in flexible electronics and photonics,” should be corrected to be “Persistent room temperature phosphorescent (RTP) materials with good mechanical properties and robust optical properties have great potential in flexible electronics and photonics.”.

Response: We appreciate the reviewer’s comments. According to the suggestion, we have revised the original sentence to “Persistent room temperature phosphorescent (RTP) materials with unique mechanical properties and robust optical properties have great potential in flexible electronics and photonics” in the revised manuscript.

4.2 In the second paragraph of “Introduction”, “The constant mechanical deformations in practical applications make those materials cannot meet the requirements for real-world usage.” should be corrected to be “The constant mechanical deformations in practical applications make those materials dissatisfy the requirements for real-world usage.”

Response: We appreciate the reviewer’s comments. According to the suggestion, we have revised the original sentence to “The constant mechanical deformations in practical applications make those materials dissatisfy the requirements for real-world usage” in the revised manuscript.

4.3 In the third paragraph of “Introduction”, “Alternatively, by blending amorphous RTP copolymers into elastomer matrices might be an effective way to obtain stretchable films with stable persistent luminescence properties.” should be corrected to be “Alternatively, blending amorphous RTP copolymers into elastomer matrices might be an effective way to obtain stretchable films with stable persistent luminescence properties.”

Response: We appreciate the reviewer’s comments. According to the suggestion, we have revised the original sentence to “Alternatively, blending amorphous RTP copolymers into elastomer matrices might be an effective way to obtain stretchable

films with stable persistent luminescence properties” in the revised manuscript.

Reviewer #3 (Remarks to the Author):

In this manuscript, the authors report a series of stretchable and multicolor persistent RTP elastomers, they exhibited high optical transparency under daylight and bright persistent luminescence after the removal of 365 nm excitation. Even after large-area and multiple times mechanical deformations, the optical properties can be well retained, which is previously unprecedented. This is an interesting work, the demonstrations are well designed, and the results are solid. It can provide important information for developing RTP elastomers, and it will be of interest to a wide readership. Based on the rational material preparation and the outstanding optical and mechanical performance, it can be recommended for publication in Nature Communications after minor revision.

Response: We would like to thank the precious time devoted to the reviewing process of the reviewer. We appreciate the referee for the positive evaluation of our manuscript. We will try our best to improve the manuscript according to the reviewer's suggestions.

1. In this work, only ^1H NMR and ^{13}C NMR were used to characterize the monomers. HRMS analysis of each of monomer is recommended to be used for further demonstration of their chemical structures.

Response: We appreciate the reviewer's comments. According to the suggestion, the high-resolution MS spectra of monomers M1-M6 have been determined and provided in the revised supplementary information.

Figure R9. HRMS spectra of monomers **M1**.

Figure R10. HRMS spectra of monomers **M2**.

Figure R11. HRMS spectra of monomers **M3**.

Figure R12. HRMS spectra of monomers **M4**.

Figure R13. HRMS spectra of monomers **M5**.

Figure R14. HRMS spectra of monomers **M6**.

2. UV-visible absorption spectra of P1-P6 are not provided.

Response: We appreciate the reviewer's comments. According to the suggestion, we have conducted further investigations on the UV absorption spectra of P1-P6 polymer films. All the polymers exhibit intense absorption peaks below 500 nm, which are attributed to intramolecular π - π , π - π^* or n - π^* transitions, as depicted in Figure R15. These additional data have been added in the revised supplementary information.

Figure R15. The UV absorption spectra of polymers P1 to P6.

3. Why 1/50 was chosen for polymerization?

Response: We appreciate the reviewer's comments. The copolymerization of M3 and acrylamide was conducted at various molar ratios, including 1:400, 1:200, 1:100, 1:50, and 1:10, resulting in a series of persistent P3 RTP polymers. As illustrated in Figure R16a, the phosphorescence peaks for these polymers are almost identical. Upon evaluation of emission decay times, it was observed that the 1:50 molar ratio resulted in the most substantial RTP lifetime of 1090 ms (Figure R16b). Consequently, for all polymers incorporated in this study, a polymerization process using a 1:50 molar ratio was adopted.

Figure R16. a The phosphorescence spectra of P3 with different ratios (1/10-1/400) under 365 nm excitation. **b** Emission lifetime decay curves of P3 (1/10-1/400) in the solid state under 365 nm excitation.

4. *How does the authors confirm the long-lasting luminescence of P1-P6 from triplet state rather singlet state?*

Response: We appreciate the reviewer's comments. To confirm the long-lasting luminescence of these polymers originated from triplet state rather than singlet state, we conducted a detailed analysis of the temperature dependence of phosphorescence in a P3 sample (1/50). As depicted in Figure R17a, over the range of 77 to 300 K, a noticeable decrease in emission intensity was observed with an increase in temperature. Further, the lifetime decay curves of P3 decreased from 1276 ms to 759 ms as the temperature increased from 77 K to 300 K. These results derived from temperature-dependent phosphorescence spectra and lifetime decay curves substantiate the conclusion that the long-lasting luminescence originates from the triplet state rather than the singlet state.

Figure R17. a Temperature dependence (77 K-300 K) phosphorescence spectra of P3 (1/50) under 365 nm excitation. **b** Temperature dependence (77 K-300 K) emission lifetime decay curves of P3 (1/50) in the solid state under 365 nm excitation.

5. *How the quantum yields were determined in this work?*

Response: We appreciate the reviewer's comments. According to the suggestion, the calculation formulas on photoluminescence quantum efficiency have been added in the revised supplementary information.

The method to calculate the fluorescence and phosphorescence quantum yields

separately was performed according to previous literatures (*Angew. Chem. Int. Ed.* **2020**, *58*, 17451-17455; *Angew. Chem. Int. Ed.* **2020**, *59*, 16054-16060). The phosphorescence bands could be obtained in their delayed emission spectrum. According to the structure of phosphorescence bands, the fluorescence and phosphorescence emission bands could be separated in steady state emission spectra. The ratio for fluorescence and phosphorescence quantum yields could be calculated with areas of separated fluorescence and phosphorescence bands. Thus, the fluorescence and phosphorescence quantum yields could be obtained with their total luminescence quantum yields and the ratio for the two relative quantum yields. The calculation method was added in the revised supplementary information.

Photoluminescence quantum efficiency was determined by using Edinburgh FLS980 spectrometer with the integrating sphere (142 mm in diameter) under ambient condition, the fluorescence and phosphorescence quantum efficiency (Φ_F and Φ_P) were calculated through the following formulas:

$$\Phi_P = \Phi_E \times \frac{A_P}{A_E} \quad (1)$$

$$\Phi_F = \Phi_E - \Phi_P \quad (2)$$

where Φ_E refers to the measured total emission quantum efficiency, A_P and A_E refer to the integral areas of phosphorescence and photoluminescence components in photoluminescence spectra, respectively.

6. *In Supplementary Tab.3, I think the P1-P6 should be E1-E6?*

Response: We appreciate the reviewer's comments. You are indeed correct, there appears to have been a typographical error in Supplementary Table 3. The labels should be E1-E6 instead of P1-P6. We apologize for the confusion caused and have rectified this error in the revised supplementary materials.

REVIEWER COMMENTS

Reviewer #1 (Remarks to the Author):

In this revised manuscript, the authors have carefully answered all the questions point-to-point from the reviewers. However, an uncertain point needs to be explained, that's mechanism diagram for grafting of PDMS and PVA without initiator. This is a good work and I, therefore, recommend publication.

Reviewer #2 (Remarks to the Author):

The revised version can be accepted.

Reviewer #3 (Remarks to the Author):

These issues I raised have been addressed by authors in revision.

Point-by-point response to the reviewers:

Reviewer #1 (Remarks to the Author):

In this revised manuscript, the authors have carefully answered all the questions point-to-point from the reviewers. However, uncertain point needs to be explain, that's mechanism diagram for grafting of PDMS and PVA without initiator. This is a good work and I, therefore, recommend publication.

Response: We would like to thank the precious time devoted to the reviewing process of the reviewer. We appreciate the reviewer's constructive comments and suggestions. Chemical modification has long been accepted as an effective approach to adjust surface property, by introducing active functional groups onto PDMS surface, such as -OH, -COOH and -NH₂. Chemical modification has two main methods, including plasma grafting and surface modification. PVA is widely used to blend with other polymers to improve their hydrophilicity and biocompatibility (*Surf. Coat. Tech.* **2012**, 206, 2161; *Sci. Rep.* **2018**, 8, 16038). In our work, we employ the plasma grafting method to improve the hydrophilicity of PDMS. As shown in Figure R1, with the increase of 365 nm irradiation time, PDMS generates more active groups and active sites to bind more PVA, which raise the hydrophilicity of PDMS. Next, P3 polymers are embedded into PDMS-PVA chains through hydrogen bonding, van-der-Walls, ion-dipole, and dipole-dipole interactions. Finally, the mixture was transferred into a fluoridated mold, and elastomer films were obtained after cooling.

Figure R1 Mechanism diagram for grafting of PDMS and PVA.

Reference

1. Li, J. Y. et al. Chemical modification on top of nanotopography to enhance surface properties of PDMS. *Surf. Coat. Tech.* **206**, 2161 (2012).
2. Perween, S. et al. PVA-PDMS-stearic acid composite nanofibrous mats with improved mechanical behavior for selective filtering applications. *Sci. Rep.* **8**, 16038 (2018).

REVIEWERS' COMMENTS

Reviewer #1 (Remarks to the Author):

This revised manuscript can be accepted.